# Effects of Nox4 upregulation on PECAM-1 expression in a mouse model of diabetic retinopathy

Jinli Wang[1], Daniel A. Lai[1], Joshua J. Wang[1], Sarah X. Zhang[1,2]*

**1** Department of Ophthalmology and Ross Eye Institute, Jacobs School of Medicine and Biomedical Sciences, University at Buffalo, State University of New York, Buffalo, NY, United States of America, **2** Department of Biochemistry, Jacobs School of Medicine and Biomedical Sciences, University at Buffalo, State University of New York, Buffalo, NY, United States of America

☯ These authors contributed equally to this work.

\* xzhang38@buffalo.edu

**Data Availability Statement:** All relevant data are within the manuscript and its Supporting Information files.

**Funding:** This research was supported by NIH/NEI Grants EY019949, EY025061, EY030970, a

## Abstract

Diabetic Retinopathy (DR) is the leading cause of vision loss in working-age adults. The hallmark features of DR include vascular leakage, capillary loss, retinal ischemia, and aberrant neovascularization. Although the pathophysiology is not fully understood, accumulating evidence supports elevated reactive oxygen species associated with increased activity of NADPH oxidase 4 (Nox4) as major drivers of disease progression. Previously, we have shown that Nox4 upregulation in retinal endothelial cells by diabetes leads to increased vascular leakage by an unknown mechanism. Platelet endothelial cell adhesion molecule 1 (PECAM-1) is a cell surface molecule that is highly expressed in endothelial cells and regulates endothelial barrier function. In the present study, using endothelial cell-specific human Nox4 transgenic (TG) mice and endothelial cell-specific Nox4 conditional knockout (cKO) mice, we investigated the impact of Nox4 upregulation on PECAM-1 expression in mouse retinas and brain microvascular endothelial cells (BMECs). Additionally, cultured human retinal endothelial cells (HRECs) transduced with adenovirus overexpressing human Nox4 were used in the study. We found that overexpression of Nox4 increases PECAM-1 mRNA but has no effect on its protein expression in the mouse retina, BMECs, or HRECs. Furthermore, PECAM-1 mRNA and protein expression was unchanged in BMECs isolated from cKO mice compared to wild type (WT) mice with or without 2 months of diabetes. Together, these findings do not support a significant role of Nox4 in the regulation of PECAM-1 expression in the diabetic retina and endothelial cells. Further studies are warranted to elucidate the mechanism of Nox4-induced vascular leakage by investigating other intercellular junctional proteins in endothelial cells and their implications in the pathophysiology of diabetic retinopathy.

research grant NGR G2019302 from the
BrightFocus Foundation, and an Unrestricted Grant
from Research to Prevent Blindness, awarded to
the Department of Ophthalmology at the State
University of New York at Buffalo. D.A.L. gratefully
acknowledges the receipt of a Dean's Summer
Research Fellowship from the Jacobs School of
Medicine and Biomedical Sciences at the State
University of New York at Buffalo. The funders had
no role in study design, data collection and
analysis, decision to publish, or preparation of the
manuscript.

**Competing interests:** The authors have declared
that no competing interests exist.

## Introduction

Diabetic Retinopathy (DR) is a common neurovascular complication of diabetes and the leading cause of vision loss in working-age adults [1]. DR is characterized by progressive damage to retinal blood vessels causing vascular leakage, vessel occlusion, extensive capillary loss, and aberrant neovascularization [2]. One major pathological change in the diabetic retina is the breakdown of the blood-retinal barrier (BRB), which leads to leakage of fluid and blood content into surrounding retinal tissue causing macular edema. Eventually, irreversible damage to the retinal vessels induces retinal ischemia which exacerbates diabetes-induced neuronal dysfunction and cell death, causing blindness [3,4]. Despite its prevalence, the pathogenesis of DR is not fully understood.

Oxidative stress is a major contributor to the pathogenesis of DR [5]. Increased reactive oxygen species (ROS), such as hydrogen peroxide, in the diabetic retina promote vascular permeability [6], leukostatic plugging [7], retinal inflammation, and pathological angiogenesis [8], which are hallmark features of DR [2]. The family of NADPH oxidase enzymes plays a critical role in diabetes-induced oxidative stress [9]. NADPH oxidase 4 (Nox4) is the most predominant isoform of Nox in retinal endothelial cells [10]. Previously, we have shown that Nox4 activity is increased in the diabetic retina and its expression is significantly upregulated by hyperglycemia and hypoxia in retinal endothelial cells [3]. Treatment with NADPH oxidase inhibitor apocynin [7] or downregulation of Nox4 using adenovirus-delivered siRNA [3] significantly reduced retinal vascular permeability in diabetic mice. Furthermore, our recent study shows that endothelial cell (EC)-specific deletion of Nox4 prevents diabetes-induced vascular leakage and acellular capillary formation in the retina [11]. These findings strongly suggest that Nox4 plays a vital role in vascular damage in DR. However, the mechanism by which Nox4 regulates vascular permeability is not fully understood.

The integrity of the endothelial barrier, which forms the inner BRB, is essential for maintaining the homeostatic environment of the neural retina. Disruption of the endothelial barrier can cause vascular hyperpermeability and leakage that leads to macular edema, retinal inflammation, retinal ischemia, and cell death. One specific surface molecule, platelet endothelial cell adhesion molecule 1 (PECAM-1), has been shown to decrease in the diabetic retina [4,12,13]. PECAM-1 is a cell surface antigen expressed on granulocytes, monocytes, platelets, and endothelial cells (ECs). It is a versatile protein with important roles in platelet function, signal transduction, and trans-endothelial migration and inflammation [14]. Previous studies have shown that PECAM-1 helps maintain the endothelial barrier [14] and vascular integrity [15,16] by tethering the adjacent EC membranes via formation of homodimers. This arrangement regulates fluid exchange between the blood and the surrounding tissue while retaining erythrocytes and large proteins within blood vessels. Other adhesion molecules that comprise tight junctions and adherens junctions also contribute to the integrity of the endothelial barrier [14,17]. It is unclear if the elevated oxidative stress resulting from Nox4 upregulation in DR directly affects PECAM-1 expression, and therefore disrupts the structural integrity of the inner BRB.

In the present study, we seek to understand if increased Nox4 expression in ECs contributes to BRB breakdown through regulation of PECAM-1. Using EC-specific hNox4 transgenic (TG) mice that overexpress human Nox4, we investigated the effect of Nox4 overexpression on PECAM-1 expression in retinal vasculature. In addition, we isolated brain microvascular endothelial cells (BMECs) from TG mice and EC-specific Nox4 conditional knockout (cKO) mice and evaluated the role of Nox4 in the regulation of PECAM-1 expression in ECs. Our results show that Nox4 upregulation does not influence PECAM-1 expression at EC junctions. Therefore, Nox4-induced vascular leakage in DR is likely mediated by the regulation of other junctional proteins.

## Materials and methods

### Animals

Generation of EC-specific human Nox4 transgenic (TG) mice and EC-specific Nox4 conditional knockout (cKO) mice were described elsewhere [11,18]. In all experiments, both male and female mice were included and balanced across experimental groups. Littermates were used as control. Mice were euthanized by exposure to CO2, followed by cervical dislocations to ensure death. All animal procedures were approved by the Institutional Animal Care and Use Committee (IACUC) at the University at Buffalo, State University of New York, and were compliant with all guidelines set forth by the Association for Research in Vision and Ophthalmology Statement for the Use of Animals in Ophthalmic and Vision Research.

### Induction of diabetes

Adult wild-type (WT) and Nox4 cKO mice received daily intraperitoneal injections of 50 mg/kg streptozotocin (STZ, Sigma Aldrich) for 5 consecutive days. Measurements of blood glucose greater than 13.9 mmol/L at 1 week after the last injection were considered diabetic. Mice were sacrificed at 2 months after induction of diabetes for cell isolation and downstream analysis.

### Cell culture

Human retinal endothelial cells (HRECs) were cultured with EGM™-2 basal medium supplemented with EGM™-2 MV Microvascular Endothelial Cell Growth Medium SingleQuots™ supplements (CC-3202, Lonza) and 1x Penicillin-Streptomycin-Glutamine (10378016, Thermofisher). HRECs were transduced with adenovirus overexpressing human Nox4 (Ad-Nox4) at a multiplicity of infection (MOI) of 25 or 50, and adenovirus with GFP (Ad-GFP) was used as control. Cells within passage 10 were used for this study. Primary mouse brain microvascular endothelial cells (BMECs) were isolated and cultured following the protocol as published previously [11,18]. Briefly, mouse brain tissue was minced into debris and digested with collagenase type I (Worthington, LS004196) / DMEM solution for 1h at 37˚C. After filtering through a 70 μm cell strainer and centrifuging in 20% bovine serum albumin (BSA) (Sigma, 2930-100GM)/DMEM containing buffer, cells were seeded in the rat collagen I (Cultrex, Catalog No. 3440-100-01)-coated plate for culturing until confluency.

### Quantitative real-time reverse transcription PCR (qRT-PCR)

Total RNA was isolated using Trizol (Life Technologies, Carlsbad, CA, USA) as previously described [18]. cDNA was synthesized with the iScript cDNA Synthesis Kits (Bio-rad, #1708891), and real-time RT-PCR was performed using the iQ™ SYBR® Green Supermix (Bio-Rad, #1708882). Primers that were used are mouse PECAM-1 forward: 5'-GTGGTCATCGCCACCTTAATA-3', reverse: 5'-TTCCACACTAGGCTCAGAAA-3'; human PECAM-1 forward: 5'-CTGAGGGTGAAGGTGATAGC-3', reverse: 5'-AGTATTTTGCTTCTGGGGAC-3'). Levels of the target genes were normalized with mouse 18S rRNA forward: 5'-GTAACCCGTTGAACCCCATT-3', reverse: 5'-CCATCCAATCGGTAGTAGCG-3'; human 18S forward: 5'-GAGGTAGTGACGAAAAATAACAAT-3', reverse: 5'-TTGCCCTCCAATGGATCCT-3').

### Western blot analysis

Mouse retinas, BMECs, and HRECs were lysed in radio immune precipitation assay (RIPA) buffer with protease inhibitor mixture, PMSF, and sodium orthovanadate (Santa Cruz Biotechnology, Santa Cruz, CA) [18]. Proteins were resolved by SDS-PAGE and then blotted with anti-PECAM-1 (R&D Systems, #AF3628), anti-Nox4 antibody (Novus Biologicals, #58851), or

anti-β-actin antibody (Sigma-Aldrich, St. Louis, MO, #1978). After incubation with HRP-conjugated secondary antibodies, membranes were developed with Clarity™ Western ECL Substrate (Bio-rad, #170–5060) using ChemiDoc MP Imaging System (Bio-Rad, Hercules, CA). The protein bands were semi-quantified via densitometry using Image J software.

## Immunohistochemistry on retinal cryosections

Mouse eyes were fixed in 4% PFA at room temperature for 45 minutes, and then washed several times with PBS. Eyes were submerged in PBS with 30% sucrose at 4˚C overnight. Afterward, tissue was embedded in an OCT block and semi-submerged in dry ice. Once frozen solid, OCT-embedded eyes were stored at -80˚C. Tissue cryosections were cut to 12 microns onto StatLab Superfrost slides. For staining, sections were blocked in Perm/Block solution (PBS + 1% Triton + 1% BSA) and incubated with primary antibody, anti-mouse PECAM-1 (1:10) (DSHB, #2H8) in Perm/Block solution, followed by incubation with secondary antibody, Alexa Fluor 488-conjugated goat anti-mouse antibody (1:800) (Invitrogen, #A11001), combined with Alexa Fluor 594-conjugated isolectin GS B4 (1:200) (Invitrogen, #I21413) in Perm/Block solution. Sections were mounted with VectaShield and immunofluorescence was examined under Olympus Provis AX-70 microscope.

## Immunofluorescence staining of retinal wholemounts

Immunostaining protocol was adapted from a published protocol [19]. Mouse eyes were enucleated and fixed in 4% PFA at room temperature for 15 minutes, then transferred to PBS for 10 minutes. After whole eye fixation, retinas were dissected and fixed with cold methanol (-20˚C) for 20 minutes, followed by blocking in Perm/Block solution (PBS + 0.3% Triton + 0.5% BSA + 5% goat serum). Goat anti-mouse PECAM-1 (1:50) (R&D Systems, Fischer Scientific, #AF3628) combined with Alexa Fluor 594-conjugated isolectin GS B4 (1:200) (Invitrogen, #I21413) in Perm/Block solution was used for primary antibody incubation, followed by Alexa Fluor 488-conjugated donkey anti-goat antibody (1:1000) (Invitrogen, #A32814) for secondary antibody incubation. After extensive washing, retinas were mounted using ProLong Glass Antifade (Invitrogen) onto a coverslip and allowed to set overnight. Immunofluorescence was captured via Olympus Provis AX-70 and Leica TCS SP8 Confocal microscopes. The fluorescence intensity was quantified using ImageJ software.

## Quantification of PECAM-1 levels in immunofluorescence images of retinal wholemounts

For image data captured with the Leica TCS SP8 confocal microscope, five to six image stacks were randomly taken in the mid-peripheral region in each retinal wholemount (582 microns x 582 microns, variable z-axis). Stacks were merged into composite images and pixel values were summated. Because the depth of the superficial retinal vasculature was not uniform across the wholemount, some image stacks captured multiple vascular layers within the field of view at the same imaging depth. To increase the accuracy of vessel analysis, smaller stacks were obtained from within the original image stack areas to isolate the superficial layer of retinal vasculature. Composite images were converted into masks using ImageJ software. Image masks were superimposed onto the original composite images to eliminate background fluorescence to zero. The levels of PECAM-1 were quantified by measuring raw integrated density of PECAM-1 fluorescence in ImageJ and dividing by total vessel length, measured with Angiogenesis Analyzer ImageJ software.

For image data collected with the Olympus Provis AX-70 motorized fluorescence microscope, five images of the superficial retinal vasculature were randomly taken two to three optic

disc lengths away from the center of the flat-mount retina at 20x magnification. Images were converted into masks using ImageJ software. Image masks were superimposed onto the original composite images to eliminate background fluorescence to zero. PECAM-1 intensity was quantified by measuring raw integrated density in ImageJ and dividing by total vessel length measured with the Angiogenesis Analyzer software. PECAM-1 intensity data derived from confocal and motorized fluorescence microscopy was combined to create a larger sample size for analysis (n = 2 mice/group before, n = 4 mice/group after). PECAM-1 intensity values were averaged for each mouse, and fold change from WT mouse retinas was calculated.

### Quantitative analysis of retinal vasculature

The total lengths for IB4-stained retinal vasculature were measured in each merged stack (confocal microscopy) or image (motorized fluorescence microscopy) to normalize PECAM-1 fluorescence and account for random differences in vascular density. The Angiogenesis Analyzer software in ImageJ was used to calculate total vessel lengths of the imaged retinal vessels.

For image data obtained via confocal microscopy, quantification was performed after manually eliminating background fluorescence from deeper vascular layers. Retinal blood vessels with heterogeneous IB4 staining were omitted from both PECAM-1 and IB4 immunofluorescence channel images due to poor software accuracy when measuring vessel lengths. Most vessels with heterogeneous IB4 staining consisted of large superficial vessels. To calculate vessel lengths for images obtained via motorized fluorescence microscopy (Olympus Provis AX-70), image masks were created and edited to remove background fluorescence. Masks were used for analysis due to poor Angiogenesis Analyzer software fidelity with original images.

### Immunofluorescent staining of BMECs

The confluent BMECs were fixed with 4% PFA at room temperature for 15 minutes. After washing, cells were blocked in PBS containing 1% Triton-X and 1% BSA at room temperature for 1h. BMECs were incubated with anti-mouse PECAM-1 antibody (1:10) (DSHB, #2H8) in the blocking solution for 1h at room temperature, followed by secondary antibody intubation. Nuclei were stained with DAPI, and images were taken by Olympus Provis AX-70 microscope.

### Statistical analysis

Statistical analyses were performed using Prism software (Version 9.5.1) as previously described [18]. Data were expressed as mean ± SD. Unpaired, two-tailed student's $t$ test was used when two experimental groups were involved. One-way analysis of variance (ANOVA) was performed when comparing three or more groups. $P < 0.05$ was considered statistically significant.

## Results

### Nox4 overexpression increases PECAM-1 mRNA in the mouse retina

To determine whether increased Nox4 expression in ECs regulates PECAM-1 in the mouse retina, we examined the mRNA and protein expression of PECAM-1 in the retina of adult WT and Nox4 TG mice. We first isolated retinal micro-vessels (RMVs) from these mice and measured PECAM-1 mRNA and protein levels by qRT-PCR and western blot analysis, respectively. We found a significant increase in Pecam-1 mRNA in RMVs from TG mice compared to WT controls (Fig 1A). However, western blot analysis showed no difference in the PECAM-1 protein level between the two groups (Fig 1B & 1C). To confirm these findings, we performed immunofluorescence staining to examine PECAM-1 expression in retinal

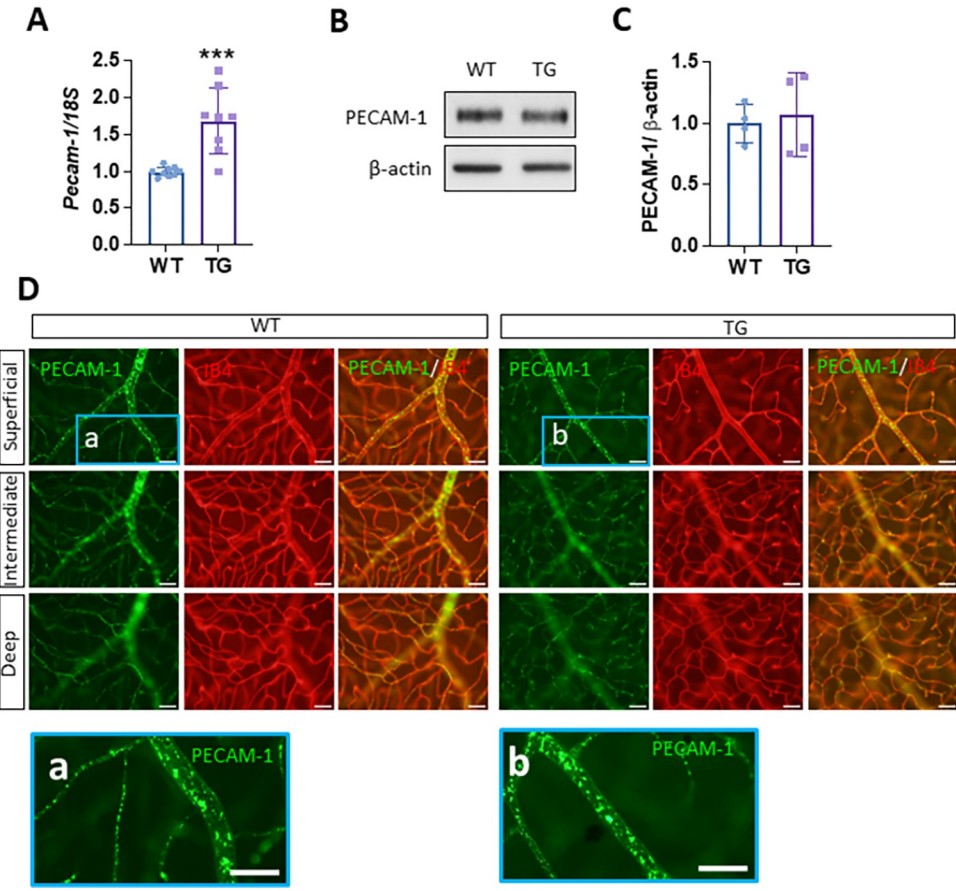

**Fig 1. Overexpression of Nox4 increases *Pecam-1* mRNA but does not affect PECAM-1 protein level in the mouse retina.** (**A**) qRT-PCR analysis of *Pecam-1* mRNA in the retinal micro-vessels (RMVs) derived from WT and TG mice aged 2–6 months (n = 8–9 mice/group, mean ± SD, ***$p<0.001$). (**B**) Western blot and (**C**) densitometry quantification of PECAM-1 protein in the retina of WT and TG mice aged 3–6 months (n = 4 mice/group, mean ± SD, $p>0.05$). (**D**) Immunofluorescence staining of PECAM-1 in retinal wholemounts from WT and TG mice aged 6–7 months using anti-PECAM-1 antibody and Isolectin B4 (IB4). Scale bars = 50 μm.

wholemounts. Our results showed robust PECAM-1 protein expression exclusively in retinal vessels, distributed into superficial, intermediate, and deep vascular layers (Fig 1D). The staining pattern of PECAM-1 in retinal arterioles and capillaries appeared discontinuous and in patches, consistent with their function as cell junctional proteins (Fig 1D, a&b).

To determine if there was a quantitative difference in PECAM-1 expression and distribution in retinal vessels in TG and WT mice, we used confocal microscopy to capture high-resolution images, which were used to measure the intensity of PECAM-1 fluorescence signal in vessels (Fig 2A). We found no difference in PECAM-1 intensity localized to the superficial vascular layer from WT and TG mouse retinas (Fig 2B & 2C). The results were further confirmed by immunohistochemistry for PECAM-1 in retinal cryosections, which demonstrated vessel-specific distribution of PECAM-1 in the retina with no qualitative difference between the two groups (Fig 2D).

## Nox4 overexpression does not affect PECAM-1 expression in BMECs

To investigate how Nox4 overexpression regulates PECAM-1 expression in ECs, we isolated BMECs from WT and TG mice at ages of 6–7 months and beyond 1 year. PECAM-1 expression was examined by immunostaining in confluent BMECs. As shown in Fig 3A, PECAM-1

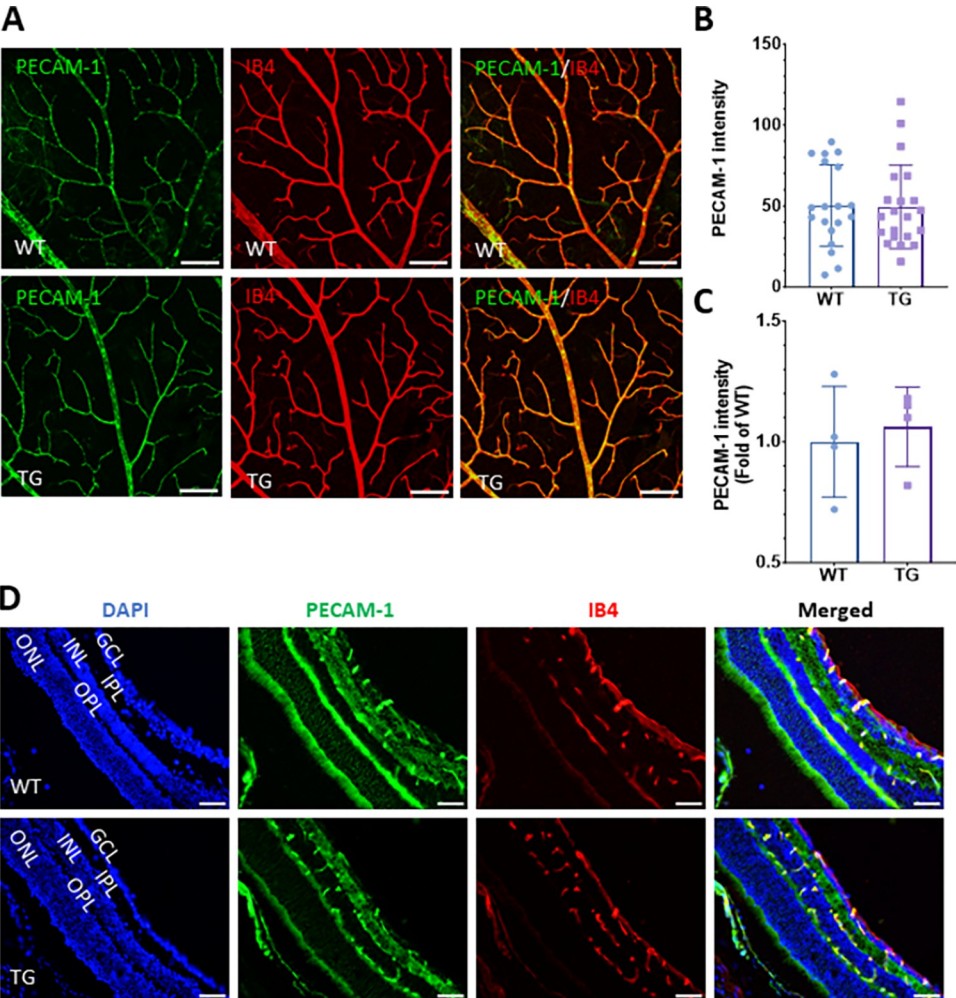

**Fig 2. Overexpression of Nox4 does not alter PECAM-1 protein level in the retinal vasculature.** PECAM-1 protein levels were determined by immunofluorescence staining in retinal wholemounts from WT and TG mice aged 6–7 months. **(A)** Representative images obtained via confocal microscopy showing the superficial vascular layer of the mouse retina with colocalization of PECAM-1 and IB4-stained vessels. Scale bars = 100 μm. **(B)** Quantification of PECAM-1 protein level in WT and TG mouse retinas using fluorescence intensity as a surrogate (derived from raw integrated density divided by total vessel length) (n = 2 mice/group, mean ± SD). **(C)** Relative fluorescence intensity of PECAM-1 protein in retinal vessels of WT and TG mice (n = 4 mice/group, mean ± SD, p>0.05). Images were captured via a combination of confocal microscopy and Olympus fluorescence microscopy. **(D)** PECAM-1 (green) and IB4 (red) staining using retinal cryosections derived from WT and TG mice. Nuclei were stained with DAPI (blue). Scale bars = 50 μm.

protein was predominantly localized to the cell surface. There were no obvious differences in PECAM-1 protein expression and distribution between WT and TG groups (Fig 3A). To further confirm the results, we performed western blot analysis to determine PECAM-1 protein levels. We found that there was a slight increase in PECAM-1 protein in BMECs derived from TG mice compared to WT mice at ages of 6–7 months (Fig 3B & 3C) or 12–16 months (Fig 3D & 3E). However, the differences did not reach significance.

## Nox4 overexpression increases PECAM-1 mRNA in HRECs

To further validate the changes in retinal ECs, we transduced HRECs with adenovirus expressing Nox4 (Ad-Nox4) or GFP (Ad-GFP) as control at a multiplicity of infection (MOI) of 25 or

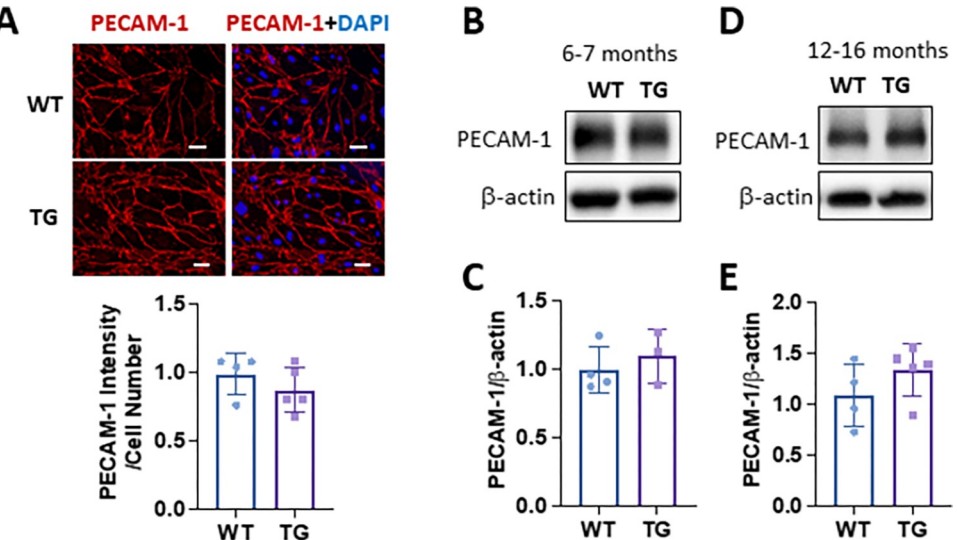

**Fig 3. Nox4 overexpression does not alter PECAM-1 protein expression in BMECs.** (**A**) Representative immunofluorescent images and quantification of BMECs using anti-PECAM-1 antibody (red). Nuclei were stained with DAPI (blue). Scale bar = 50 μm, n = 4–5 mice/group, mean ± SD, p>0.05. (**B&D**) Western blot and (**C&E**) densitometry quantification of PECAM-1 protein expression in BMECs isolated from mice at ages of 6–7 months (3–4 mice/group, mean ± SD, p>0.05) (**B&C**) or 12–16 months (4–5 mice/group, mean ± SD, p>0.05) (**D-E**).

50. At both doses, Ad-Nox4 induced a significant increase in Nox4 expression in HRECs. PECAM-1 mRNA and protein levels were determined by qRT-PCR and western blot analysis, respectively. Our results show that Nox4 overexpression at the lower dose (MOI: 25) did not affect PECAM-1 expression, but the higher dose (MOI: 50) induced a significant increase in PECAM-1 mRNA level (Fig 4A). At the protein level, Nox4 overexpression at the lower dose (MOI: 25) failed to alter PECAM-1 expression; however, the higher dose (MOI: 50) showed a modest effect of reducing PECAM-1 expression, although the change did not reach statistical significance (Fig 4B & 4C).

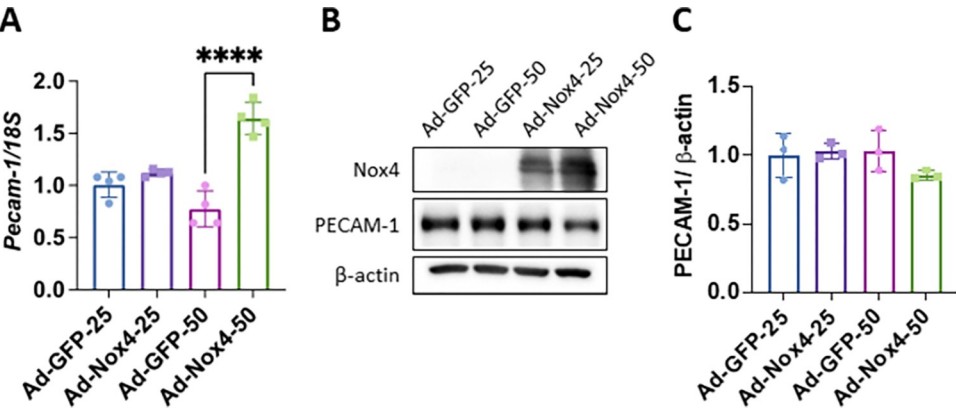

**Fig 4. Nox4 overexpression increased PECAM-1 mRNA but not protein levels in HRECs.** (**A**) qRT-PCR analysis of PECAM-1 mRNA in HRECs transduced with adenoviruses expressing Nox4 or GFP protein at a multiplicity of infection (MOI) of 25 or 50 (n = 4, mean ± SD, **** p<0.0001). (**B**) Western blot and (**C**) densitometry quantification of PECAM-1 protein in Ad-Nox4 or Ad-GFP transduced HRECs (n = 3, mean ± SD, p>0.05).

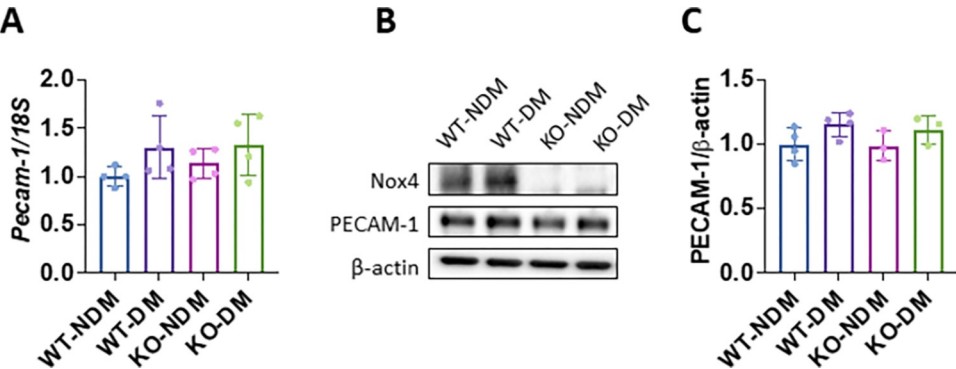

**Fig 5. Knockout of Nox4 does not alter PECAM-1 expression in BMECs under diabetes conditions.** (**A**) qRT-PCR analysis of *Pecam-1* mRNA expression in BMECs isolated from non-diabetic (NDM) or diabetic (DM) WT and cKO mice after 2 months of diabetes. (n = 4 mice/group, mean ± SD, p>0.05). (**B**) Western blot and (**C**) densitometry quantification of PECAM-1 protein levels in BMECs (n = 3–4 mice/group, mean ± SD, p>0.05).

## Knockout of Nox4 does not change PECAM-1 expression in diabetic BMECs

Previously we have shown that Nox4 upregulation contributes to BRB breakdown and vascular leakage in the diabetic retina [11]. To investigate if Nox4 regulates PECAM-1 expression in diabetic conditions, we induced diabetes in cKO mice and isolated BMECs from non-diabetic and diabetic cKO and WT mice after 2 months of diabetes. Western blot analysis showed a significant reduction of Nox4 expression in BMECs derived from cKO mice, confirming the knockout efficiency of Nox4 in ECs. In addition, we confirmed an increase of Nox4 expression in the diabetic WT ECs compared to non-diabetic WT cells. To determine the changes of PECAM-1, we performed qRT-PCR and western blot analysis. We found slight increases in both mRNA (Fig 5A) and protein (Fig 5B & 5C) levels of PECAM-1 in diabetic WT BMECs compared to non-diabetic WT controls; however, the differences are not statistically significant. Similarly, we found no changes in PECAM-1 mRNA and protein levels between WT and cKO mouse BMECs regardless of the presence of diabetes (Fig 5A–5C).

## Discussion

Vascular hyperpermeability is a key feature of diabetic vascular damage resulting in macular edema and vision impairment in DR. Although the underlying process is not completely elucidated, one principal mechanism involves disruption of the cell-to-cell junctions at the BRB. Using humanized Nox4 transgenic mice that overexpress human Nox4 in ECs, conditional knockout mice that deplete the endogenous Nox4 gene in ECs, and cultured primary BMECs and HRECs, we sought to determine if Nox4 regulates the expression of EC junction protein PECAM-1. We found that Nox4 upregulation induced an increase in PECAM-1 mRNA expression in mouse retinas and HRECs transduced with a high dose of Nox4 adenovirus. However, the protein levels of PECAM-1 were not changed by overexpression of Nox4 or deletion of the Nox4 gene in mouse retinas or BMECs. Intriguingly, overexpression of Nox4 at a higher dose resulted in a modest, not significant, reduction of PECAM-1 protein level in HRECs. This suggests that the increased PECAM-1 mRNA expression in the TG retina and HRECs could be a compensatory response to damaged BRB or reduced PECAM-1 protein level in these conditions.

Accumulating evidence suggests destruction of endothelial junction proteins contributes to vascular hyperpermeability in DR, but the topic remains controversial. Recent studies found

that PECAM-1 is reduced in rat retinal microvascular endothelial cells (RRMECs) exposed to high glucose conditions and in retinas of STZ-induced diabetic rats [4,20,21], with a rise of ubiquitinated PECAM-1 in high glucose conditions [21]. Furthermore, knockdown of *Pecam-1* by siRNA increases endothelial permeability in RRMECs cultured in high glucose condition [20]. These findings suggest that loss of PECAM-1 likely contributes to vascular leakage in DR. Contradictory to these observations, other studies have shown no change in PECAM-1 expression in the retina of diabetic mice [22,23]. For example, Sakaue and associates [22] found no change in PECAM-1 immunofluorescence signal in retinal vasculature between prediabetic state and 9 months after diabetes in a STZ mouse model. Furthermore, they found that loss of claudin-5, another major tight junction protein, correlated with vascular leakage in the retina. These observations suggest that loss of claudin-5 may be an important factor in endothelial barrier dysfunction and vascular leakage in DR, at least in some animal models. Our results corroborate these findings and do not suggest a change of PECAM-1 protein level in ECs during early diabetes. Future studies will investigate whether Nox4 induces endothelial barrier damage through dysregulation of tight junction proteins such as claudin-5.

Although we found the PECAM-1 protein level to be unchanged in the diabetic ECs, this does not exclude a potential implication of PECAM-1 in DR. One recent study has shown that matrix metalloproteinase 2 (MMP2) is upregulated in the retinas and RRMECs of diabetic rats, and the researchers suggested MMP2 may directly destroy PECAM-1 either via proteolysis or shedding [4]. Several studies have shown that peroxynitrite, a powerful oxidizing molecule, is increased in mouse and rat diabetic retinas and retinal endothelial cells under high glucose conditions [24–30]. Peroxynitrite can activate pro-MMPs without cleaving off their inhibitory domains [4,31–33], promoting MMP2-induced PECAM-1 loss at the protein level. This mechanism may indicate that PECAM-1 expression is regulated by different mechanisms other than Nox4 in diabetic retinas. The lack of change in PECAM-1 protein level in the retina or ECs of diabetic mice at 2 months or 9 months of diabetes, as observed in this study and by others [22,23], may suggest discrepancies in mouse and rat DR models. Alternatively, PECAM-1 expression at additional time points after the onset of diabetes should be investigated in future studies.

Our study has some limitations. First, the quantification of PECAM-1 immunofluorescence was performed only on the superficial vascular layer of retinal wholemounts. This may have prevented us from looking at the potential quantitative differences in PECAM-1 levels at the intermediate and deep vascular layers, although we have not observed any qualitative difference across the groups. Second, as discussed above, we did not investigate PECAM-1 changes in diabetic mice beyond 2 months of diabetes. An investigation using mice with a longer period of diabetes would have strengthened our conclusion. Nevertheless, increased vascular leakage has been previously reported to occur before 2 months of diabetes [34,35]. PECAM-1 level was unchanged in the retina at 2 months of diabetes. This does not support a strong association between PECAM-1 loss and vascular permeability in the retina in early diabetes.

In summary, our results do not imply a significant role of Nox4 in the regulation of PECAM-1 expression in DR. Given the pivotal role of NADPH oxidases [9,36], in particular Nox4 [3,18], in ROS generation in retinal ECs, understanding how Nox4 upregulation and increased ROS mediate diabetes-induced vascular damage [37,38] would have high translational significance. Indeed, a recent study demonstrates that topical administration of a Nox4 inhibitor successfully reduced retinal inflammation, neurodegeneration, and vascular leakage [39]. Future investigation to elucidate the role of Nox4 in the regulation of endothelial tight junction proteins, such as claudin-5, could provide important insight into the mechanisms of Nox4 inhibition on protecting retinal vessels in DR.

## Supporting information

**S1 File.**
(PDF)

## Acknowledgments

The authors thank Wade J Sigurdson, PhD for technical support in acquisition of confocal microscopic images and analysis at the Jacobs School of Medicine and Biomedical Sciences.

## Author Contributions

**Conceptualization:** Jinli Wang, Daniel A. Lai, Joshua J. Wang, Sarah X. Zhang.

**Data curation:** Jinli Wang, Daniel A. Lai.

**Formal analysis:** Jinli Wang, Daniel A. Lai.

**Investigation:** Jinli Wang, Daniel A. Lai, Joshua J. Wang, Sarah X. Zhang.

**Methodology:** Jinli Wang, Daniel A. Lai, Joshua J. Wang, Sarah X. Zhang.

**Project administration:** Joshua J. Wang, Sarah X. Zhang.

**Resources:** Sarah X. Zhang.

**Supervision:** Joshua J. Wang, Sarah X. Zhang.

**Validation:** Jinli Wang, Daniel A. Lai, Joshua J. Wang, Sarah X. Zhang.

**Visualization:** Jinli Wang, Daniel A. Lai, Joshua J. Wang, Sarah X. Zhang.

**Writing – original draft:** Jinli Wang, Daniel A. Lai.

**Writing – review & editing:** Jinli Wang, Daniel A. Lai, Joshua J. Wang, Sarah X. Zhang.

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
