## [Decision Letter · Decision Letter 0]

5 Dec 2023

PONE-D-23-36248Minimal Effects of Nox4 Upregulation on PECAM-1 Expression in

Diabetic RetinopathyPLOS ONE

Dear Dr. Zhang,

Thank you for submitting your manuscript to PLOS ONE. After careful consideration, we feel that it has merit but does not fully meet PLOS ONE’s publication criteria as it currently stands. Therefore, we invite you to submit a revised version of the manuscript that addresses the points raised during the review process.

Thank you for submitting the following manuscript to PLOS ONE. Please revise the manuscript according to the reviewers' comments and upload the revised file.

We look forward to receiving your revised manuscript.

Kind regards,

Yung-Hsiang Chen, Ph.D.

Academic Editor

PLOS ONE

Journal Requirements:

3. To comply with PLOS ONE submissions requirements, in your Methods section, please provide additional information regarding the experiments involving animals and ensure you have included details on (1) methods of sacrifice, (2) methods of anesthesia and/or analgesia, and (3) efforts to alleviate suffering.

"This work was supported by NIH/NEI Grants EY019949, EY025061, EY030970, research grant NGR G2019302 from the BrightFocus Foundation, and an Unrestricted Grant from Research to Prevent Blindness to the Department of Ophthalmology, State University of New York at Buffalo."

5. Thank you for stating the following in the Acknowledgments Section of your manuscript: "This work was supported by NIH/NEI Grants EY019949, EY025061, EY030970, research grant NGR G2019302 from the BrightFocus Foundation, and an Unrestricted Grant from Research to Prevent Blindness to the Department of Ophthalmology, State University of New York at Buffalo."

 "This work was supported by NIH/NEI Grants EY019949, EY025061, EY030970, research grant NGR G2019302 from the BrightFocus Foundation, and an Unrestricted Grant from Research to Prevent Blindness to the Department of Ophthalmology, State University of New York at Buffalo."

"REVISED/ACCEPT

Additional Editor Comments :

Thank you for submitting the following manuscript to PLOS ONE.

Please revise the manuscript according to the reviewers' comments and upload the revised file.

Reviewers' comments:

Reviewer's Responses to Questions

**Comments to the Author**

1. Is the manuscript technically sound, and do the data support the conclusions?

Reviewer #1: Partly

Reviewer #2: Partly

Reviewer #3: Yes

2. Has the statistical analysis been performed appropriately and rigorously? 

Reviewer #1: Yes

Reviewer #2: Yes

Reviewer #3: Yes

3. Have the authors made all data underlying the findings in their manuscript fully available?

Reviewer #1: Yes

Reviewer #2: Yes

Reviewer #3: Yes

4. Is the manuscript presented in an intelligible fashion and written in standard English?

Reviewer #1: Yes

Reviewer #2: Yes

Reviewer #3: Yes

5. Review Comments to the Author

Reviewer #1: In this work, Nox4 was studied, and it seemed that it could not affect the mRNA and protein levels of PECAM-1 in DR mice and associated cells. After reading the whole manuscript, there are several confusing points.

1. In the authors' previous work, diabetes brought the upregulation of Nox4, which could lead to the increasing of retinal vascular leakage. And in this work, PECAM-1 was selected as a candidate, however, how was it screened? Was there any supporting information? The direct interaction between Nox4 and PECAM-1? Are there any intermediate steps？

2. As PECAM-1 could not be regulated by Nox4, what's the exact downstream target of PECAM-1?

3. The human and mice Nox4s shared relatively high sequence identity, however, there might be some epitopes. Why the human Nox4 transgenic mice was used? If the authors hope to investigate the effects of Nox4 overexpression in mice, the mice Nox4 overexpression plasmid or AAV should be used.

Reviewer #2: The figures of this work are appropriate. However, several points would make readers confused.

1. The result of this work is Nox4 could not regulate PECAM-1 in DR. So, why Nox4 and PECAM-1 were investigated? What could be affected by Nox4 in the progression of DR?

2. Are there any intermediate factors between these two molecules?

3. Why a human Nox4 transgenic mice model was used? Although Nox4 is conserved in human and mice, their sequences are not same. If the authors hope to investigate the effects of Nox4 overexpression and knockdown in mice, the mice Nox4 should be overexpressed but not human Nox4.

Reviewer #3: Wang et al investigate the impact of Nox4 upregulation on PECAM-1 expression in mouse retinas and brain microvascular endothelial cells (BMECs) using endothelial cell-specific human Nox4 transgenic (TG) mice and endothelial cell-specific Nox4 conditional knockout (cKO) mice. The manuscript is well written, well presented, and presents convincing evidence of not support a significant role of Nox4 in the regulation of PECAM-1 expression in the diabetic retina and endothelial cells. However, there are some major problems need to be addressed.

Major points:

1.The breakdown of endothelial junction proteins, such as ZO-1, occludin, and claudin, has been shown to contribute to vascular hyperpermeability in DR. This study, however, aims to investigate the role of Nox4 in relation to PECAM-1. It would be helpful to know if there is any preliminary data, such as RNA-sequencing data of endothelial cells with NOX4 interference, that could support this investigation.

2.Platelet endothelial cell adhesion molecule 1 (PECAM-1) is a cell surface molecule that is highly expressed in endothelial cells, but also expressed in various circulating cells, such as platelets, monocytes, neutrophils, and certain T cells. In this study, was vascular perfusion performed in animal experiment to eliminate the interference of the blood cells?

3.In figure 1 and figure 2, please point out the age of the TG mice.

4.In figure 1 and figure 2, this study was lack of the result of retinal vascular permeability of TG mice. There was no difference in PECAM-1 expression between WT and TG mouse retinas in this study. Is this possibly related to the absence of retinal vascular leakage in this particular group of TG mice?

5.In figure 2D, is PECAM-1 specific to EC? In retinal cryosections derived from WT and TG mice, the PECAM-1(green) also appears to be expressed in other tissues.

6.In figure 1 and figure 2, quantitative intensity analysis of PECAM-1 / IB4 ratio in the immunofluorescence staining of retinal wholemounts is suggested.

7.Please show the quantitative intensity analysis of PECAM-1 in figure 3A, It appears that the intercellular junctions in the TG group are thicker compared to the WT group.

8.PECAM-1 has been shown to decrease in the diabetic retina. However, the slight increases in both mRNA (Fig. 5A) and protein (Fig. 5B&5C) levels of PECAM-1 in diabetic WT BMECs were observed compared to non diabetic WT controls in the present study. Please explain the inconsistency.

6. PLOS authors have the option to publish the peer review history of their article (what does this mean?). If published, this will include your full peer review and any attached files.

Reviewer #1: **Yes: **Yu Xin

Reviewer #2: **Yes: **Jun Shao

Reviewer #3: No

---

## [Author Response · Author response to Decision Letter 0]

15 Dec 2023

Dear Reviewers,

Thank you so much for carefully evaluating our manuscript and providing constructive comments to improve our manuscript. Following your suggestions, we have revised the manuscript and provided additional information/analyses as requested. We feel that these changes have greatly strengthened the manuscript. Below we provide point-to-point responses to each of your comments and summarize the changes we made to the manuscript. 

Reviewer #1

1. In the authors' previous work, diabetes brought the upregulation of Nox4, which could lead to the increasing of retinal vascular leakage. And in this work, PECAM-1 was selected as a candidate, however, how was it screened? Was there any supporting information? The direct interaction between Nox4 and PECAM-1? Are there any intermediate steps？

Response: Thank you for your comments. As discussed in the manuscript (Introduction and Discussion sections), PECAM-1 is a cell adhesion molecule highly expressed in cell-cell junctions of endothelial cells. It regulates cell junction integrity and thus contributes to endothelial barrier function. Previous studies demonstrated that PECAM-1 is downregulated in the diabetic retina and downregulation of PECAM-1 contributes to increased vascular permeability in diabetic retinopathy. Our prior work has shown that Nox4 upregulation in endothelial cells leads to increased vascular leakage. Therefore, in this study, we sought to determine whether Nox4’s effect on vascular leakage was through PECAM-1 downregulation. To our knowledge, there is no published study showing a direct interaction between PECAM-1 and Nox4. Our results from the current study do not support a regulatory role of Nox4 in PECAM-1 expression in retinal endothelial cells during diabetes. Our ongoing studies are investigating the effects of Nox4 on regulations of other junctional complexes such as tight junctions in retinal endothelial cells and DR. 

2. As PECAM-1 could not be regulated by Nox4, what's the exact downstream target of PECAM-1?

Response: The downstream target and upstream regulators of PECAM-1 have been investigated in several recent studies (see Discussion section). In diabetic retinopathy, increased peroxynitrite activates MMP2, which interacts with PECAM-1 protein resulting in its degradation (Eshaq RS and Harris NR, IOVS, 2019;60:748-60; cited as Ref. 4). Hyperglycemia and inflammatory factors (TNF-α and IFN-γ) can also reduce PECAM-1 protein level in retinal endothelial cells by increasing its ubiquitination and proteasomal degradation (Eshaq RS and Harris NR, Microcirculation. 2020;27:e12596; Eshaq RS and Harris NR, Microcirculation. 2021;28:e12717; cited as Refs. 20 and 21). As to the downstream target, the authors showed that PECAM-1 can interact with β-catenin and loss of PECAM-1 leads to increased vascular permeability through reducing β-catenin levels in retinal endothelial cells (Eshaq RS and Harris NR, Microcirculation. 2021;28:e12717; Ref. 20). In addition, PECAM-1 itself is an adhesion junctional protein regulating endothelial cell junctional integrity.

3. The human and mice Nox4s shared relatively high sequence identity, however, there might be some epitopes. Why the human Nox4 transgenic mice was used? If the authors hope to investigate the effects of Nox4 overexpression in mice, the mice Nox4 overexpression plasmid or AAV should be used.

Response: Thank you for your comments. The humanized Nox4 transgenic mouse line was generated in our previous study (Tang, Xixiang, et al. "Sustained Upregulation of Endothelial Nox4 Mediates Retinal Vascular Pathology in Type 1 Diabetes." Diabetes, 2023, 72: 112-125; cited as Ref. 11). Human Nox4 overexpression was used because of its relevance to human disease of DR. In this study, we have shown that Nox4 overexpression induces endothelial cell dysfunction, focal vascular leakage, and capillary degeneration, which recapitulate the major vascular pathology of DR. Given our goal is to investigate the role of Nox4 in endothelial cells in DR, overexpression of Nox4 by plasmid or AAV via intravitreal injection is not feasible, because of the poor transfection/transduction efficiency of retinal endothelial cells. 

Reviewer #2

1. The result of this work is Nox4 could not regulate PECAM-1 in DR. So, why Nox4 and PECAM-1 were investigated? What could be affected by Nox4 in the progression of DR?

Response: Please see responses to Reviewer#1, Comment 1, and response to Reviewer #1, Comment 3. 

2. Are there any intermediate factors between these two molecules?

Response: Please see responses to Reviewer#1, Comment 1. 

3. Why a human Nox4 transgenic mice model was used? Although Nox4 is conserved in human and mice, their sequences are not same. If the authors hope to investigate the effects of Nox4 overexpression and knockdown in mice, the mice Nox4 should be overexpressed but not human Nox4.

Response: Please see responses to Reviewer#1, Comment 3. 

Reviewer #3

Major points:

1.The breakdown of endothelial junction proteins, such as ZO-1, occludin, and claudin, has been shown to contribute to vascular hyperpermeability in DR. This study, however, aims to investigate the role of Nox4 in relation to PECAM-1. It would be helpful to know if there is any preliminary data, such as RNA-sequencing data of endothelial cells with NOX4 interference, that could support this investigation.

Response: Thank you for the comment. We did not perform an RNA-sequencing study of endothelial cells with Nox4 interference. As you mentioned, the breakdown of endothelial tight junction proteins contributes to vascular permeability in DR. Loss of PECAM-1 has also been shown to increase vascular permeability (Ref. 20). We feel that the regulation of PECAM-1 is less intensively studied compared to tight junction proteins such as ZO-1, occludin, and claudin. Thus, in the present study, we determined whether Nox4 upregulation alters PECAM-1 expression in endothelial cells. Our ongoing studies are investigating the effects of Nox4 on the regulation of other junctional complexes focusing on tight junction proteins in retinal endothelial cells and DR.

2.Platelet endothelial cell adhesion molecule 1 (PECAM-1) is a cell surface molecule that is highly expressed in endothelial cells, but also expressed in various circulating cells, such as platelets, monocytes, neutrophils, and certain T cells. In this study, was vascular perfusion performed in animal experiment to eliminate the interference of the blood cells?

Response: Thank you for the valid point. We did not perform vascular perfusion in the animal experiment. We used isolectin B4 to label endothelial cells in retinal vessels and observed PECAM-1 well colocalized with isolectin B4, suggesting its localization to endothelial cells. To further explore the regulation of PECAM-1 in endothelial cells by Nox4, we isolated and cultured the brain microvascular endothelial cells (BMECs) from TG mice. Thus, we believe that these measures have helped in excluding the confounding factors from the circulating cells. 

3.In figure 1 and figure 2, please point out the age of the TG mice.

Response: We have included the age information for Fig. 1 and Fig. 2 in the figure legends.

4.In figure 1 and figure 2, this study was lack of the result of retinal vascular permeability of TG mice. There was no difference in PECAM-1 expression between WT and TG mouse retinas in this study. Is this possibly related to the absence of retinal vascular leakage in this particular group of TG mice?

Response: Please see the response to Reviewer #1, Comment 3. In a previous study (Ref. 11), we demonstrated increased vascular leakage and capillary degeneration in the TG mice. 

5. In figure 2D, is PECAM-1 specific to EC? In retinal cryosections derived from WT and TG mice, the PECAM-1(green) also appears to be expressed in other tissues.

Response: Thank you for the comment. As mentioned in the Introduction, PECAM-1 is highly expressed at endothelial cell junctions but is also present on the surface of other circulating cells including granulocytes, monocytes, and platelets. In our experiments using immunohistochemistry of retinal cryosections, we observed nonspecific background fluorescence of anti-PECAM-1 antibody outside of IB4-stained vasculature. However, it is noteworthy that PECAM-1 immunostaining in the retinal whole mounts did not exhibit significant background fluorescence. This observation supports that the increased background signal is likely unrelated to extravascular PECAM-1 but rather caused by an unknown factor intrinsic to the process of preparing the cryosection samples. 

6. In figure 1 and figure 2, quantitative intensity analysis of PECAM-1 / IB4 ratio in the immunofluorescence staining of retinal wholemounts is suggested.

Response: Thank you for the valuable suggestion. In our analysis of PECAM-1 immunofluorescence staining in retinal whole mounts, we employed IB4 staining as a qualitative control to demonstrate the colocalization of PECAM-1 within the retinal vasculature. Thus, we did not undertake a quantitative analysis of IB4 fluorescence. IB4 was used as a marker of endothelial cells. Therefore, we respectfully believe it would not be beneficial to use its fluorescence as a quantitative control value in the form of PECAM-1 / IB4 ratio. 

7.Please show the quantitative intensity analysis of PECAM-1 in figure 3A, It appears that the intercellular junctions in the TG group are thicker compared to the WT group.

Response: Thank you for the suggestion. We have added the quantitative intensity analysis of PECAM-1 for Fig. 3A. The result showed no statistical differences in PECAM-1 intensity between WT and TG groups (n= 4-5 mice/group, t-test, P>0.05).

8. PECAM-1 has been shown to decrease in the diabetic retina. However, the slight increases in both mRNA (Fig. 5A) and protein (Fig. 5B&5C) levels of PECAM-1 in diabetic WT BMECs were observed compared to non diabetic WT controls in the present study. Please explain the inconsistency.

Response: We have discussed extensively the inconsistency in PECAM-1 changes in the diabetic retina, reported in the present and previous studies. Please see the Discussion section, lines 324 – 363.

---

## [Decision Letter · Decision Letter 1]

11 Jan 2024

PONE-D-23-36248R1Minimal Effects of Nox4 Upregulation on PECAM-1 Expression in Diabetic RetinopathyPLOS ONE

Dear Dr. Zhang,

Thank you for submitting your manuscript to PLOS ONE. After careful consideration, we feel that it has merit but does not fully meet PLOS ONE’s publication criteria as it currently stands. Therefore, we invite you to submit a revised version of the manuscript that addresses the points raised during the review process.

Thank you for submitting the following manuscript to PLOS ONE.

Please revise the manuscript according to the reviewers' comments and upload the revised file.

We look forward to receiving your revised manuscript.

Kind regards,

Yung-Hsiang Chen, Ph.D.

Academic Editor

PLOS ONE

Journal Requirements:

Additional Editor Comments:

Thank you for submitting the following manuscript to PLOS ONE.

Please revise the manuscript according to the reviewers' comments and upload the revised file.

Reviewers' comments:

Reviewer's Responses to Questions

**Comments to the Author**

1. If the authors have adequately addressed your comments raised in a previous round of review and you feel that this manuscript is now acceptable for publication, you may indicate that here to bypass the “Comments to the Author” section, enter your conflict of interest statement in the “Confidential to Editor” section, and submit your "Accept" recommendation.

Reviewer #1: All comments have been addressed

Reviewer #2: (No Response)

Reviewer #3: All comments have been addressed

2. Is the manuscript technically sound, and do the data support the conclusions?

Reviewer #1: Partly

Reviewer #2: Partly

Reviewer #3: (No Response)

3. Has the statistical analysis been performed appropriately and rigorously? 

Reviewer #1: N/A

Reviewer #2: Yes

Reviewer #3: Yes

4. Have the authors made all data underlying the findings in their manuscript fully available?

Reviewer #1: Yes

Reviewer #2: (No Response)

Reviewer #3: Yes

5. Is the manuscript presented in an intelligible fashion and written in standard English?

Reviewer #1: Yes

Reviewer #2: Yes

Reviewer #3: Yes

6. Review Comments to the Author

Reviewer #1: Most of the comments have been answered or discussed. Two points should be carefully considered.

1. The title revealed a negative result that Nox4 upregulation could not significantly affect the expression of PECAM-1 in DR. Could the title be reconstructed in a more appropriated sentence?

2. Using human Nox 4 mice is a serious problem, although the sequence identity between human and mouse Nox4 was ~90.8%, human Nox4 is still a foreigner and potential antigen in mouse body, which might bring with undesirable immunological reactions. In the authors' response, DR is a human disease and human Nox4 was used. However, the DR process was built in a mice model, it's also a mice disease.

Reviewer #2: Part or the review comments have been answered and revisions have been made.

But the most significant problem is still the expression of human Nox4 gene in mice mode, which has been also queried by reviewer #1. The author's answer is that "Human Nox4 overexpression was used because of its relevance to human disease of DR. In this study, we have shown that Nox4 overexpression induces endothelial cell dysfunction, focal vascular leakage, and capillary degeneration, which recapitulate the major vascular pathology of DR." There are several problems.

1. DR is a human disease, however, in this work, the whole DR progression was in mice model, using a human element in a whole mice progression is not appropriated.

2. Although human and mice Nox4 show homologous sequence identity, there are still some fragments which could be recognized as epitopes and may lead to immunological reactions. In the clinical diagnosis and fundamental research of DR, local inflammation is also one of the important features.

Reviewer #3: (No Response)

7. PLOS authors have the option to publish the peer review history of their article (what does this mean?). If published, this will include your full peer review and any attached files.

Reviewer #1: **Yes: **Yu Xin

Reviewer #2: **Yes: **Jun Shao

Reviewer #3: No

---

## [Author Response · Author response to Decision Letter 1]

8 Feb 2024

Reviewer 1, Question #1: The title revealed a negative result that Nox4 upregulation could not significantly affect the expression of PECAM-1 in DR. Could the title be reconstructed in a more appropriate sentence?

Response to Reviewer: Thank you very much for this insightful suggestion. We re-formulate the title and change it to “Effects of Nox4 Upregulation on PECAM-1 Expression in a Mouse Model of Diabetic Retinopathy”. We believe that it is more suitable and concise now.

Reviewer 1, Question #2: Using human Nox 4 mice is a serious problem, although the sequence identity between human and mouse Nox4 was ~90.8%, human Nox4 is still a foreigner and potential antigen in mouse body, which might bring with undesirable immunological reactions. In the authors' response, DR is a human disease and human Nox4 was used. However, the DR process was built in a mice model, it's also a mice disease.

Response to Reviewer: We totally understand and appreciate your concern regarding the use of human Nox4 transgenic mice. To address your question, we have consulted experts in genetic animal models and have searched the literature extensively in PubMed and Google Scholar. Transgenic mice overexpressing human genes or cells have been valuable pre-clinical models to study human diseases for over 30 years. No report demonstrates undesirable immunological reactions to human gene in transgenic mice. This is because once a human gene is inserted into the mouse genomic locus, it is considered as an integral part of the mouse genome. The protein expressed in the mouse tissue will be considered as its own protein, which will not cause an immune response. Please refer to Ref. 1 and Ref. 2 for detailed information on how B cell- and T cell-mediated self-tolerance is established. As to the human Nox transgenic mice, human Nox4 is expressed in Tie2-expressing cells at the embryonic stage. In addition, Nox4 is an intracellular protein that is not exposed to the innate immune system. Therefore, we do not expect an immune response caused by human Nox4 in the transgenic mice. 

Reviewer 2, Question #1: DR is a human disease, however, in this work, the whole DR progression was in mice model, using a human element in a whole mice progression is not appropriated.

Response to Reviewer: We understand your concern and thank you for comments. Animal models are widely employed in scientific research and are essential tools for the study of most, if not all, human diseases. Due to logistical and ethical restrictions to study human genes in human subjects, genetically humanized mice have become increasingly important as animal models for the study of human diseases (Ref. 3). Like most proteins, Nox4 is not 100% conserved between human and mouse. Specifically, there is a ~90.8% sequence homology between human and mouse Nox4. The differences in these orthologous sequences may have functional consequences (Ref. 4). In addition, human-specific genes and the encoded proteins are likely to function as regulators of cell signaling events specific to human diseases (Ref. 5). Therefore, the use of humanized NOX4 mice is an appropriate and valuable model in studying human disease like DR.

Reviewer 2, Question #2: Although human and mice Nox4 show homologous sequence identity, there are still some fragments which could be recognized as epitopes and may lead to immunological reactions. In the clinical diagnosis and fundamental research of DR, local inflammation is also one of the important features.

Response to Reviewer: We totally understand and appreciate your concern. Please refer to Response to Reviewer 1, Question #2.

References

1. Nemazee D. Mechanisms of central tolerance for B cells. Nat Rev Immunol. 2017;17(5):281-294. doi: 10.1038/nri.2017.19. PMID: 28368006.

2. Xing Y, Hogquist KA. T-cell tolerance: central and peripheral. Cold Spring Harb Perspect Biol. 2012;4(6):a006957. doi: 10.1101/cshperspect.a006957. PMID: 22661634.

3. Walsh NC, Kenney LL, Jangalwe S, Aryee KE, Greiner DL, Brehm MA, Shultz LD. Humanized Mouse Models of Clinical Disease. Annu Rev Pathol. 2017; 12:187-215. doi: 10.1146/annurev-pathol-052016-100332. PMID: 27959627.

4. Devoy A, Bunton-Stasyshyn RK, Tybulewicz VL, Smith AJ, Fisher EM. Genomically humanized mice: technologies and promises. Nat Rev Genet. 2011;13:14-20. doi:10.1038/nrg3116

5. Stahl PD, Wainszelbaum MJ. Human-specific genes may offer a unique window into human cell signaling. Sci Signal. 2009;2(89):pe59. doi: 10.1126/scisignal.289pe59. PMID: 19797272.

---

## [Decision Letter · Decision Letter 2]

14 Feb 2024

PONE-D-23-36248R2Effects of Nox4 Upregulation on PECAM-1 Expression in a Mouse Model of Diabetic RetinopathyPLOS ONE

Dear Dr. Zhang,

Thank you for submitting your manuscript to PLOS ONE. After careful consideration, we feel that it has merit but does not fully meet PLOS ONE’s publication criteria as it currently stands. Therefore, we invite you to submit a revised version of the manuscript that addresses the points raised during the review process.

Thank you for submitting the following manuscript to PLOS ONE.

Please carefully revise the manuscript according to the reviewers' comments and upload the revised file.

We look forward to receiving your revised manuscript.

Kind regards,

Yung-Hsiang Chen, Ph.D.

Academic Editor

PLOS ONE

Journal Requirements:

Additional Editor Comments:

Thank you for submitting the following manuscript to PLOS ONE.

Please carefully revise the manuscript according to the reviewers' comments and upload the revised file.

Reviewers' comments:

Reviewer's Responses to Questions

**Comments to the Author**

1. If the authors have adequately addressed your comments raised in a previous round of review and you feel that this manuscript is now acceptable for publication, you may indicate that here to bypass the “Comments to the Author” section, enter your conflict of interest statement in the “Confidential to Editor” section, and submit your "Accept" recommendation.

Reviewer #1: All comments have been addressed

Reviewer #2: (No Response)

2. Is the manuscript technically sound, and do the data support the conclusions?

Reviewer #1: Partly

Reviewer #2: Partly

3. Has the statistical analysis been performed appropriately and rigorously? 

Reviewer #1: N/A

Reviewer #2: (No Response)

4. Have the authors made all data underlying the findings in their manuscript fully available?

Reviewer #1: (No Response)

Reviewer #2: (No Response)

5. Is the manuscript presented in an intelligible fashion and written in standard English?

Reviewer #1: (No Response)

Reviewer #2: (No Response)

6. Review Comments to the Author

Reviewer #1: Not all the self-proteins are safe, immunology is a complicated system. A great deal of autoimmunity progressions and associated diseases have been reported. The human Nox4 protein might bind with some mice proteins to form epitopes. I have attended a number of clinical trials of protein drugs, and in animal work, the proteins/antibodies targeting human proteins usually don't work well in mice/rats/rabbits/monkeys.

Although transgenic animals have been used in the research of human diseases; the animal molecules work in the progression of a certain disease in animal models. My suggestion is to try the knock down/knock out/overexpression of mice Nox4 in mice model.

If the authors do not want to design the experiments of mice Nox4, the title should be clarified as "Effects of Human Nox4 Upregulation on PECAM-1 Expression in a Mouse Model of Diabetic Retinopathy". Obviously, human Nox4 could not displace mice Nox4 in associated signal axes; and then this work seems to make no sense.

Reviewer #2: After carefully reading all the comments and responses, I think I agree with another reviewer's comments. Human body and mouse are two different machines; human and mice Nox4 are different components in these two machines, although these two molecules shared high similarity, they are not the same. The conclusion of this manuscript is that overexpression of Nox4 increases PECAM-1 mRNA but has no effect on its protein expression in the mouse retina, BMECs, or HRECs. Could the authors try to knock down or overexpress mice Nox4 in eye in mice model using AAV vector? In addition, as reported, most of the signal pathways are crosslinked, and PECAM-1 should not be the only target of Nox4, the authors should screen the direct targets of Nox4 via co-IP or other associated protocols.

7. PLOS authors have the option to publish the peer review history of their article (what does this mean?). If published, this will include your full peer review and any attached files.

Reviewer #1: **Yes: **Yu Xin

Reviewer #2: No

---

## [Author Response · Author response to Decision Letter 2]

9 Apr 2024

Reviewer 1

Comments: Not all the self-proteins are safe, immunology is a complicated system. A great deal of autoimmunity progressions and associated diseases have been reported. The human Nox4 protein might bind with some mice proteins to form epitopes. I have attended a number of clinical trials of protein drugs, and in animal work, the proteins/antibodies targeting human proteins usually don't work well in mice/rats/rabbits/monkeys.

Although transgenic animals have been used in the research of human diseases; the animal molecules work in the progression of a certain disease in animal models. My suggestion is to try the knock down/knock out/overexpression of mice Nox4 in mice model.

If the authors do not want to design the experiments of mice Nox4, the title should be clarified as "Effects of Human Nox4 Upregulation on PECAM-1 Expression in a Mouse Model of Diabetic Retinopathy". Obviously, human Nox4 could not displace mice Nox4 in associated signal axes; and then this work seems to make no sense.

Response to Reviewer: We very much appreciate the reviewer’s further comments. We agree that although humanized animal models have been widely used and are becoming a powerful model in biomedical research, they do have some limitations. We also appreciate the reviewer’s constructive suggestions of “try the knock down/knock out/overexpression of mice Nox4 in mice model”. The knockout mice lacking endogenous mouse Nox4 in endothelial cells, referred to as “EC-specific Nox4 conditional knockout (cKO) mice”, were used in this study. Results from EC-Nox4 cKO mice were presented in Figure 5. 

Reviewer 2 

Comments: After carefully reading all the comments and responses, I think I agree with another reviewer's comments. Human body and mouse are two different machines; human and mice Nox4 are different components in these two machines, although these two molecules shared high similarity, they are not the same. The conclusion of this manuscript is that overexpression of Nox4 increases PECAM-1 mRNA but has no effect on its protein expression in the mouse retina, BMECs, or HRECs. Could the authors try to knock down or overexpress mice Nox4 in eye in mice model using AAV vector? In addition, as reported, most of the signal pathways are crosslinked, and PECAM-1 should not be the only target of Nox4, the authors should screen the direct targets of Nox4 via co-IP or other associated protocols.

Response to Reviewer: Thank you for taking time to carefully read all the comments and responses. We also appreciate the reviewer’s valuable suggestions--“try to knock down or overexpress mice Nox4 in eye in mice model using AAV vector and screen the direct targets of Nox4 via co-IP”. However, screening the direct targets of Nox4 is beyond the scope of the current manuscript.

---

## [Decision Letter · Decision Letter 3]

18 Apr 2024

Effects of Nox4 Upregulation on PECAM-1 Expression in a Mouse Model of Diabetic Retinopathy

PONE-D-23-36248R3

Dear Dr. Zhang,

We’re pleased to inform you that your manuscript has been judged scientifically suitable for publication and will be formally accepted for publication once it meets all outstanding technical requirements.

Kind regards,

Yung-Hsiang Chen, Ph.D.

Academic Editor

PLOS ONE

Additional Editor Comments (optional):

Congratulations on the acceptance of your manuscript, and thank you for your interest in submitting your work to PLOS ONE.

Reviewers' comments:

Reviewer's Responses to Questions

**Comments to the Author**

1. If the authors have adequately addressed your comments raised in a previous round of review and you feel that this manuscript is now acceptable for publication, you may indicate that here to bypass the “Comments to the Author” section, enter your conflict of interest statement in the “Confidential to Editor” section, and submit your "Accept" recommendation.

Reviewer #1: All comments have been addressed

Reviewer #2: All comments have been addressed

2. Is the manuscript technically sound, and do the data support the conclusions?

Reviewer #1: Yes

Reviewer #2: Yes

3. Has the statistical analysis been performed appropriately and rigorously? 

Reviewer #1: Yes

Reviewer #2: Yes

4. Have the authors made all data underlying the findings in their manuscript fully available?

Reviewer #1: Yes

Reviewer #2: Yes

5. Is the manuscript presented in an intelligible fashion and written in standard English?

Reviewer #1: Yes

Reviewer #2: Yes

6. Review Comments to the Author

Reviewer #1: (No Response)

Reviewer #2: (No Response)

7. PLOS authors have the option to publish the peer review history of their article (what does this mean?). If published, this will include your full peer review and any attached files.

Reviewer #1: **Yes: **Yu Xin

Reviewer #2: No

---

## [Editor Report · Acceptance letter]

3 May 2024

PONE-D-23-36248R3 

PLOS ONE

Dear Dr. Zhang, 

I'm pleased to inform you that your manuscript has been deemed suitable for publication in PLOS ONE. Congratulations! Your manuscript is now being handed over to our production team.

Kind regards, 

on behalf of

Dr. Yung-Hsiang Chen 

Academic Editor

PLOS ONE